# Effects of Manure Waste Biochars in Mining Soils

**María Luisa Álvarez [1], Ana Méndez [1], Jorge Paz-Ferreiro [2] 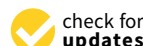 and Gabriel Gascó [3,\*]**

1   Department of Geological and Mining Engineering, Universidad Politécnica de Madrid, 28003 Madrid, Spain; marisa.alvarezc@alumnos.upm.es (M.L.Á.); anamaria.mendez@upm.es (A.M.)
2   School of Engineering, RMIT University, GPO Box 2476, Melbourne, VIC 3001, Australia; jorge.paz-ferreiro@rmit.edu.au
3   Department of Agricultural Production, Universidad Politécnica de Madrid, 28040 Madrid, Spain
\*   Correspondence: gabriel.gasco@upm.es; Tel.: +34-910671044

**Abstract:** Land degradation by old mining activities is a concern worldwide. However, many known technologies are expensive and cannot be considered for mining soil restoration. Biochar amendment of mining soils is becoming an interesting alternative to traditional technologies due to an improvement in soil properties and metal mobility reduction. Biochar effects depend on soil and biochar properties, which in turn vary with pyrolysis conversion parameters and the feedstock used. The objective of this study is to evaluate the effect of four biochars prepared from poultry and rabbit manure at two pyrolysis temperatures (450 and 600 °C) in the trace metal mobility, $CO_2$ emissions, and enzymatic activity of 10 mining soils located in three historical mining areas of Spain (Zarandas-Andalusia, Mijarojos-Cantabria, and Portman-Murcia). For this reason, soils were amended with biochars at a rate of 10% (*w/w*), and different treatments were incubated for 180 days. For acid soils of the Zarandas-Andalusia area, biochar addition reduced the mobility of Ni, Zn, Cd, Pb, and Cr, respectively, by 91%, 81%, 29%, 67%, and 70%. Nevertheless, biochars did not exhibit the same efficiency in the other two areas where alkaline soils were predominant. $CO_2$ emissions generally increased in the treated soils. The application of biochars produced at 600 °C reduced $CO_2$ emissions, in some cases by more than 28%, being an adequate strategy for C sequestration in soil. The results showed that application of manure biochars can be an effective technique to reduce the mobility of metals in multi-contaminated acid soils, while reducing metal toxicity for soil microorganisms.

**Keywords:** biochar; mobile metal content; $CO_2$ emissions; enzyme activities

## 1. Introduction

Traditionally, mineral extraction and processing for metal production are associated with landscape modifications and an important environmental risk. Metals, metalloids, organic compounds, and salts may be transferred to the soil, water, and air, affecting the trophic chain [1]. Furthermore, impacts on soil derived from mining activities, including erosion, significant pH variations, runoff, or higher metal mobility, produce soil degradation and groundwater pollution risk. In addition, the characteristics of mining soils hinder the success of establishing a vegetative cover once the mines are closed.

In order to mitigate the above-mentioned environmental impacts, new remediation and restoration techniques are required. Traditionally, soil excavation [2], leaching of pollutants [3], electro-migration [4], soil-flushing [5], or the use of barrier technology [6] are used for this purpose. Advances in the last years modified these techniques to improve them. However, most of them are still expensive and lead to total loss of soil fertility [7].

Alternative techniques, which are environmentally friendly and with low implementation costs, are gradually gaining public acceptance. The most classical and famous techniques are

phytoremediation, biotransformation, and more recently, the use of amendments [7–9], including compost, sewage sludge, crop residues, urban wastes, or manures [10–12]. Organic wastes can be used as amendments in mining soils due to their high content in nutrients and organic matter, mainly lignin and cellulose, with functional groups, including hydroxyls or carboxyls that allow complexation with metals and reduce their mobility [13]. However, the high mineralization and decomposition rates of organic wastes in soils result in low amendment stability, resulting in the need for re-applying the amendment. In addition, the potential presence of toxic compounds in wastes such as metal(loids), polychlorinated biphenyls (PCBs), or polychlorinated hydrocarbons (PHAs) are associated with risks to soil biota [14,15].

More recently, there is an increasing interest in biochar as a soil amendment [16,17]. Biochar is a carbon-rich material obtained by pyrolysis or thermal treatment of biomass under restricted $O_2$ atmosphere. During biochar production, the thermal treatment of organic wastes allows the elimination of some toxic elements present in the feedstock [18]. In addition, biochar acts as a soil fertilizer, increasing nutrient supply and plant growth [19], and as an adsorbent, reducing the bioavailability of metals [20]. The effect of biochar in soils depends greatly on the biochar characteristics, which are influenced by pyrolysis conditions and feedstock characteristics [21,22]. For this reason, an exhaustive biochar characterization is essential to understand the biochar effects in a particular soil. Biochars produced from plants show low nutrient content, which implies the use of additional fertilizers [22]. However, manure-derived biochar can be used as an organic amendment due to its high nutrient content. For example, Cely et al. [23] prepared biochar (pyrolysis temperatures: 300–500 °C) from cattle manure, poultry litter, and pig manure with 1.2–4.7%, 1.2–2.3 g·kg$^{-1}$, and 8–35% P, N, and organic carbon, respectively. Gascó et al. [24] produced biochars from pig manure, which were suitable for both agronomic and remediation purposes. In addition, previous works indicated that manure-derived biochars show higher metal adsorption capacity than plant-derived biochars [25]. Kumar et al. [26] observed a greater zinc immobilization using biochar from cattle manure than utilizing grain husk biochar, while Ebadnejad et al. [27] found cadmium and lead immobilization higher than 51% and 68.8%, respectively, with respect to control soil in two soils amended with poultry manure biochar at rates of 5–10%.

However, the reduction of the bioavailability and toxicity of metals in mining soils is not the only objective of soil remediation. Biochar addition could play an important role in improving soil physical, chemical, and biological properties. For these reasons, the main objective of the present work is to evaluate the effect of the addition of four manure-derived biochars on the chemical and biological properties of different mining soils.

## 2. Materials and Methods

### 2.1. Selected Manure Wastes and Biochar Production

Two manure wastes were selected as feedstock for biochar production, poultry litter (PL) and rabbit manure (RM), obtained from farms located in the practice field of Technical University of Madrid (Spain). Samples were air-dried, crushed, and sieved below 2 mm. Four biochars were prepared as follows: 500 g of air-dried manures (PL or RM) were introduced in a 2-L steel reactor inside a Hastelloy autoclave supplied by Demede (http://demede.es). Samples were heated at 3 °C·min$^{-1}$ until 450 °C (BPL450 and BRM450) or 600 °C (BPL600 and BRM600). In all cases, the final temperature was maintained for 1 h. $N_2$ flow of 0.5 L·min$^{-1}$ was used during thermal treatment. The steel reactor has two thermocouples; one is inserted into the recipient in contact with the sample and the other is in contact with the external part of the steel wall. The temperature difference between the two thermocouples is 10 °C. The four resulting biochars were crushed and sieved below 2 mm.

## 2.2. Selected Mining Soils

Ten mining soils were selected from three different mining areas of Spain (Figure 1): Zarandas-Andalusia, Mijarojos-Cantabria, and Portman-Murcia. Three soils were collected in the nearby area to the copper mine of Zarandas-Andalusia: a *Cambisol* Z1 (latitude: 37°40′26.6412″ north (N), longitude: 6°34′25.3205″ west (W)) and two *Technosols* Z2 (latitude: 37°40′26.2776″ N, longitude: 6°34′05.6150″ W) and Z3 (latitude: 37°40′36.8868″ N, longitude: 6°34′04.0657″ W). This area is located in the Iberian Pyrite Belt, one of the largest metallic sulfide deposits in the world with presence of Fe, Zn, Cu, Pb, Cd, As, Ag, Sn, or Au. The other three soils in the Mijarojos-Cantabria zone, a lead–zinc mine, were as follows: two *Technosols* M1 (latitude: 43°19′57.9573″ N, longitude: 4°5′2.8573″ W) and M3 (latitude: 43°20′15.9000″ N, longitude: 4°4′28.2415″ W), and a *Cambisol* M2 (latitude: 43°20′03.9876″ N, longitude: 4°4′49.5997″ W). The main minerals of this area are sphalerite, galena, pyrite, and secondary minerals of Zn and Pb. Finally, four soil were collected from another nearby area to a lead–zinc mine in Portman-Murcia: two *Cambisols* P1 (latitude: 37° 35′24.1593″ N, longitude: 0°51′39.6248″ W) and P2 (latitude: 37° 35′22.0308″ N, longitude: 0°51′54.4914″ W), and two *Technosols* P3 (latitude: 37°35′19.2552″ N, longitude: 0°51′54.6936″ W) and P4 (latitude: 37°35′23.0712″ N, longitude: 0°51′38.8381″ W). The main minerals of this area are sulfides (sphalerite, galena, pyrite), phyllosilicates, and carbonates. The sampled *Technosols* are mine tailings. All soil samples were air-dried, crushed, and sieved below 2 mm.

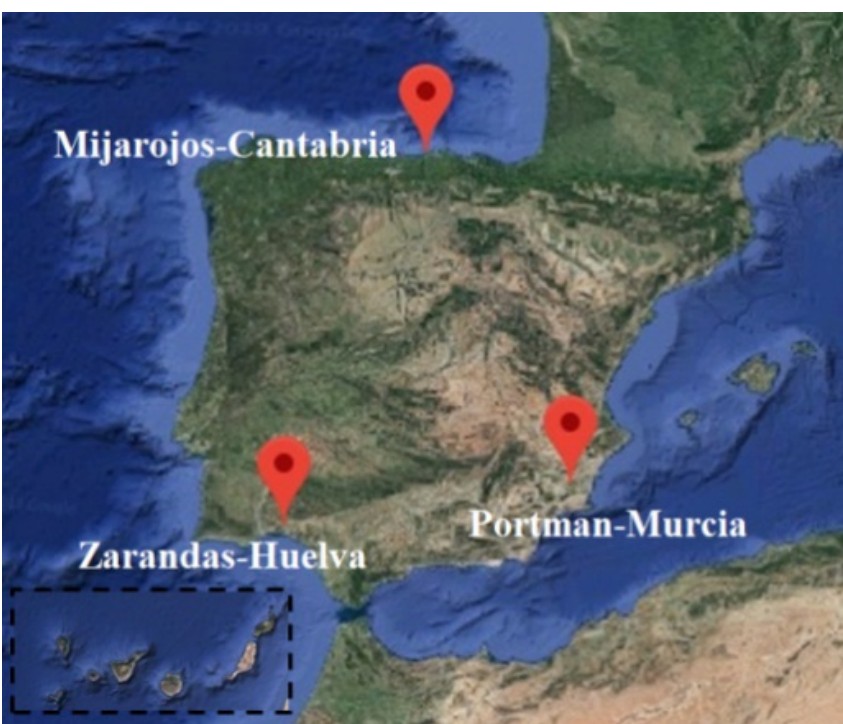

**Figure 1.** Map of Spain with the three selected mining areas.

## 2.3. Material Characterization

### 2.3.1. Raw Materials and Biochars

Manure wastes and biochars were characterized as follows: pH and electrical conductivity (EC) were determined in a sample–water ratio of 0.1:25 [28,29] using a Crison micro-pH 2000 and a Crison 222 conductivity meter, respectively. The effective cation exchange capacity (CEC) was determined using the standardized method described by ISO 23470 [30], based on centrifuge extraction with 0.0166 M cobalt(III) hexamine chloride solution. Available phosphorous (P-Olsen) was determined by the Olsen method [30], while oxidizable organic carbon (C-oxi) was determined by the Walkley–Black method [31].

The exchangeable potassium (K) was measured in the solution using a Perkin Elmer AAnalyst 400 Atomic Absorption Spectrophotometer. Soluble organic matter (SOC) was extracted by addition of 10 mL of water to 1 g (agitation 1 h) and determined using Nelson and Sommers methodology [32]. Elemental analysis was performed using an Elemental Analyzer Model Perkin-Elmer 2400 CHN. The oxygen percentage was determined by difference, according to the following equation:

$$O = 100 - (Ash + C + H + N + S)(\%), \tag{1}$$

where Ash is the ash content (%), O is the total oxygen content (%), C is the total carbon content (%), H is the total hydrogen content (%), N is the total nitrogen content (%), and S is the total sulfur content (%). Following that, O/C and H/C ratios were calculated from the elemental analysis results. Pseudo-total heavy metal concentration was determined using a Perkin Elmer 2280 atomic absorption spectrophotometer after sample digestion with 3:1 (*v/v*) concentrated $HCl/HNO_3$ following the USEPA 3051a method [33].

Biochar porosity and volume of mesopore (Vmeso) and macropore (Vmacro) were determined by Hg porosimetry, which was carried out using a Micromeritics AutoPore IV 9500 equipment. BET surface area, adsorption average pore, micropore volume (Vmicro), and micropore area (Amicro) of biochar were analyzed by nitrogen adsorption isotherm, which was carried out at 77 K in a Micromeritics Tristar 3.00.

### 2.3.2. Mining Soils

Selected soils were characterized as follows: pH and EC were determined in a sample–water ratio of 10:25 *w/v* [28,29] using a Crison micro-pH 2000 and a Crison 222 conductivity meter, respectively. The CEC, P-Olsen, K, and pseudo-total heavy metal content were determined using the same methods as in raw materials and biochars (see Section 2.3.1). Soil texture was analyzed by Bouyoucos densimeter methodology [34].

### 2.4. Incubation Experiment

In order to study the effect of biochar addition on soil properties, each soil was amended with 10% in weight of biochar (BPL450, BPL600, BRM450, or BRM600), leading to 40 soil treatments. This dose of biochar was already used in previous works [35,36]. In addition, 10 soils without biochar addition were used as control. Then, 50 g of each treatment was introduced into 1-L airtight jars. For jars with control soils, 50 g of soils were added, and, for jars with amended soils, 45 g of soil + 5 g of biochar were combined. All experiments were performed in triplicate. Later, all treatments were incubated for 180 days at constant temperature (21 ± 2 °C) and humidity (60% field capacity) in a Radiber AGP-360-HR incubator. The soil microbial respiration was measured as follows: $CO_2$ was collected in a 0.3 N NaOH solution, which was titrated using 0.3 N HCl after the $BaCl_2$-assisted precipitation of carbonates [37]. After the incubation experiment, pH, EC, mobile metal content, soil microbial biomass carbon, and enzymatic activities of soils were determined. Cu, Cd, Cr, Fe, Ni, Pb, and Ni mobile fractions were extracted with $CaCl_2$ 0.01 M, filtered and flushed with distilled water, and measured in an AAnalyst 400 PerkinElmer (AAS).

Soil metabolic quotient ($qCO_2$) and enzymatic dehydrogenase, phosphomonoesterase, and β-glucosidase activities were determined as follows: $qCO_2$ was calculated as the ratio between microbial respiration and microbial biomass C. Microbial biomass carbon was measured by the difference between the carbon content of the fumigated and unfumigated samples (following a chloroform fumigation–extraction method) with a commonly used factor (Kc = 0.45) for soils amended with biochar [38,39]. Then, $qCO_2$ was expressed as micrograms of $CO_2$–C released per milligrams of biomass carbon per hour. In the case of dehydrogenase activity, 2-(*p*-iodophenyl)-3-(*p*-nitrophenyl)-5-phenyltetrazolium chloride (INT) was used as substrate, following the method of Camiña et al. [40], and the activity of the enzyme was expressed in μmol INTF·$g^{-1}$·$h^{-1}$.

Phosphomonoesterase and β-glucosidase activities were determined after incubating soils with a substrate containing a *p*-nitrophenyl moiety; then, in the enzymatic hydrolysis, the amount of released *p*-nitrophenol was measured (μmol *p*-nitrophenol·g$^{-1}$·h$^{-1}$) [38]. Finally, to integrate information from these enzymatic activities, the geometric mean (GMea) was calculated as follows:

$$GMea = (Dehydrogenase \times \beta - Glucose \times Phosphomonoesterase)^{1/3}. \tag{2}$$

*2.5. Statistical Analysis*

The significance of the differences among means was assessed by analysis of variance (ANOVA). Duncan's multiple range test (*p* < 0.05) was implemented using the Statgraphics Centurion XVI.I. software for the calculations. Every analysis was performed in triplicate.

## 3. Results

*3.1. Material Characterization*

3.1.1. Raw Materials and Biochars

Table 1 shows the main characteristics of manure wastes, biochars, and soils. Two manure wastes and their corresponding biochars showed alkaline pH (9.01–10.85). EC values of biochars (0.40–0.50 dS·m$^{-1}$) were slightly higher than those of manure wastes (0.36 dS·m$^{-1}$ and 0.44 dS·m$^{-1}$ for RM and PL, respectively). The Coxi content of manure wastes decreased after pyrolysis, especially at the highest temperature (600 °C). The evolution of CEC during after pyrolysis greatly depends on feedstock characteristics. Biochar obtained after pyrolysis of PL showed the highest CEC values, whereas BRM600 exhibited lower CEC than RM waste. Pyrolysis decreased the P-Olsen content, particularly in biochars obtained after pyrolysis of RM (BRM450 and BRM600).

**Table 1.** Main properties of manure wastes and biochars.

| | PL | BPL450 | BPL600 | RM | BRM450 | BRM600 |
|---|---|---|---|---|---|---|
| pH | 9.01a [1] | 10.07b | 10.73c | 9.14a | 10.59c | 10.88c |
| EC (dS·m$^{-1}$) | 0.44ab | 0.47bc | 0.50c | 0.36a | 0.40a | 0.47bc |
| C-oxi (%) | 12.9c | 11.4b | 4.30a | 25.8d | 11.6bc | 4.99a |
| SOC (%) | 1.14d | 0.36b | 0.10a | 0.57c | 0.15a | 0.04a |
| CEC (mmol·kg$^{-1}$) | 89a | 123b | 131c | 139d | 151e | 132c |
| P-Olsen (mg·kg$^{-1}$) | 3214c | 2427b | 2123b | 3296c | 959a | 751a |
| K (kg·kg$^{-1}$) | 4.87a | 19.26c | 20.73d | 7.01b | 19.31c | 18.72c |
| Ash (%) | 48.26 | 61.02 | 60.23 | 41.69 | 54.44 | 62.74 |
| C (%) | 33.79 | 32.71 | 25.27 | 45.67 | 29.13 | 24.99 |
| H (%) | 4.55 | 2.70 | 0.84 | 6.17 | 1.76 | 1.22 |
| N (%) | 2.06 | 2.19 | 1.20 | 4.01 | 1.50 | 0.38 |
| O (%) | 10.87 | 0.91 | 12.10 | 1.69 | 12.79 | 10.31 |
| S (%) | 0.47 | 0.47 | 0.36 | 0.77 | 0.38 | 0.36 |
| H/C ratio (%) | 1.62 | 0.99 | 0.40 | 1.62 | 0.73 | 0.64 |
| O/C ratio (%) | 0.24 | 0.02 | 0.36 | 0.03 | 0.33 | 0.31 |
| BET surf. area (m$^2$·g$^{-1}$) | - | 4.28 | 7.03 | - | 5.68 | 35.97 |
| Adsorp. average pore width (Å) | - | 245.56 | 199 | - | 166.23 | 104.81 |
| Amicro (m$^2$·g$^{-1}$) | - | 1.14 | 2.37 | - | 1.79 | 4.20 |
| Vmicro (cm$^3$·g$^{-1}$) | - | 0.03 | 0.03 | - | 0.02 | 0.09 |
| Vmeso (cm$^3$·g$^{-1}$) | - | 0.05 | 0.07 | - | 0.04 | 0.07 |
| Vmacro (cm$^3$·g$^{-1}$) | - | 1.25 | 1.03 | - | 1.65 | 2.12 |
| Porosity (%) | - | 66.19 | 63.95 | - | 75.25 | 72.72 |
| Cd (mg·kg$^{-1}$) | 0.03a | 0.24c | 0.33d | 0.06b | 0.36e | 0.35de |
| Cr (mg·kg$^{-1}$) | 2.01b | 4.48c | 4.72c | 0.97a | 8.22d | 8.33d |
| Cu (mg·kg$^{-1}$) | 33.9a | 53.79c | 65.7d | 44.3b | 73.7e | 61.1d |
| Fe (mg·kg$^{-1}$) | 2833a | 6763c | 8715e | 5657b | 7991d | 14,916f |
| Ni (mg·kg$^{-1}$) | 2.85a | 4.63c | 5.58d | 4.34b | 6.66e | 7.93f |
| Pb (mg·kg$^{-1}$) | 7.43a | 15.4d | 13.6c | 7.95a | 9.59b | 11.0b |
| Zn (mg·kg$^{-1}$) | 468a | 546b | 701d | 623c | 870f | 753e |

[1] Values in different columns followed by the same letter are not significantly different (*p* = 0.05) using the Duncan test.

### 3.1.2. Soil Characterization

With respect to mining soil characteristics (Table 2), soils from the Zarandas area showed extremely acidic to very strong acidic pH with values between 3.63 (Z·) and 4.68 (Z1), whereas, in the Portman area, soils were slightly to moderately alkaline (pH from 7.90 in P3 to 8.16 in P1). Soils from the Mijarojos area showed moderately acidic to neutral pH values, which were between 5.67 (M1) and 7.30 (M3). P3 and P4 soils exhibited the highest EC values (1.49 and 2.24 dS·m$^{-1}$, respectively) but did not reach saline EC values (>4 dS·m$^{-1}$). P-Olsen values of soils were between 424 and 10,001 mg·kg$^{-1}$ for P3 and P1, respectively.

**Table 2.** Main properties of soils.

| | Z1 | Z2 | Z3 | P1 | P2 | P3 | P4 | M1 | M2 | M3 |
|---|---|---|---|---|---|---|---|---|---|---|
| pH | 4.68c [1] | 4.34b | 3.63a | 8.16i | 8.09h | 7.90g | 7.89g | 5.67d | 5.84e | 7.30f |
| EC (dS·m$^{-1}$) | 0.06a | 0.09a | 0.22bc | 0.19b | 0.35d | 1.49f | 2.24g | 0.49e | 0.23c | 0.25c |
| C-oxi (%) | 1.34c | 0.22a | 0.10a | 0.78b | 0.92b | 0.39a | 0.25a | 1.73d | 5.38e | 1.83d |
| CEC (mmol·kg$^{-1}$) | 11.7e | 3.64a | 2.88a | 23.9f | 12.2e | 9.90c | 10.6cd | 10.2c | 7.10b | 11.3de |
| P-Olsen (mg·kg$^{-1}$) | 1307bc | 1949c | 988ab | 10001e | 6632d | 424a | 722ab | 1915c | 706ab | 961ab |
| K (g·kg$^{-1}$) | 0.12a | 0.10a | 0.09a | 5.05c | 4.97c | 6.13d | 8.33e | 4.34b | 7.21f | 6.93e |
| Texture | Loamy | Loamy | Sandy loam | Sandy loam | Sandy clay loam | Sandy clay loam | Sandy loam | Sandy loam | Loamy sand | Sandy loam |

[1] Values in different columns followed by the same letter are not significantly different (*p* = 0.05) using the Duncan test.

Soil metal contents of selected mining soils are summarized in Table 3. Soils of the Zarandas-Andalusia area showed the lowest Cd content (<0.5 mg·kg$^{-1}$), while soil Cd content in the Portman-Murcia area and in the M1 soil of the Mijarojos-Cantabria area was higher than 5 mg·kg$^{-1}$. As content was higher than 40 mg·kg$^{-1}$ in the three studied areas, except for M2 soil (19.3 mg·kg$^{-1}$) located in the Mijarojos-Cantabria area. Both soil Ni and Cr contents were lower than 100 mg·kg$^{-1}$ in the three areas, while soil Pb content was higher than 100 mg·kg$^{-1}$ except for M2 soil in the Mijarojos-Cantabria area. With respect to Zn, the Portman-Murcia area had the highest Zn soil content (>2575 mg·kg$^{-1}$) followed by Mijarojos-Cantabria soils (180–11,035 mg·kg$^{-1}$) and the Zarandas-Andalusia area (81.4–219 mg·kg$^{-1}$). Finally, soils of the Zarandas-Andalusia area and the P2 soil of the Portman-Murcia area had soil Cu content higher than 125 mg·kg$^{-1}$, while the remaining soils had values lower than 60 mg·kg$^{-1}$.

**Table 3.** Pseudo-total metal content in soils.

| | Z1 | Z2 | Z3 | P1 | P2 | P3 | P4 | Ref. Values [1] | M1 | M2 | M3 | Ref. Values [2] | Critical Values [5] |
|---|---|---|---|---|---|---|---|---|---|---|---|---|---|
| | | | | | | **(dry mg·kg$^{-1}$, except Fe in dry g·kg$^{-1}$)** | | | | | | | |
| Cd | 0.5 | 0.5 | 0.5 | 5.84 | 20.1 | 8.62 | 16.4 | 750 | 12.7 | 0.99 | 1.81 | 1.0 | 3–8 |
| Cr | 97.3 | 2.56 | 2.42 | 61.5 | 39.2 | 50.9 | 43.2 | 10,000 [3] 100 [4] | 72.0 | 43.8 | 44.4 | 118.0 | 75–100 |
| Cu | 183 | 374 | 248 | 34.3 | 207 | 35.1 | 47.0 | 10,000 | 13.6 | 11.2 | 9.12 | 34.0 | 60–125 |
| Fe | 58.9 | 91.6 | 98.0 | 113 | 175 | 153 | 185 | - | 201 | 26.6 | 43.1 | - | - |
| Ni | 27.5 | 2.47 | 2.13 | 44.8 | 39.9 | 29.5 | 31.4 | 10,000 | 43.1 | 8.80 | 27.2 | 52.0 | 100 |
| Pb | 104 | 296 | 422 | 5072 | 2848 | 2321 | 2770 | 2750 | 3295 | 82.2 | 174 | 58.0 | 100–400 |
| Zn | 130 | 219 | 81.4 | 2575 | 8573 | 3580 | 6373 | 10,000 | 11,035 | 180 | 758 | 272.0 | 70–400 |
| As | 65.5 | 236 | 528 | 144 | 1183 | 234 | 530 | 40 | 110 | 19.3 | 44.4 | 38.0 | 20–50 |

[1] Standard limits for industrial use of soils in Andalusia were used as reference values for Zarandas-Andalusia and Portman-Murcia areas [41]. [2] Risk-Based Soil Screening Levels for Cantabria soils were used as reference values for Mijarojos-Cantabria area [42]. [3] Standard limit value for Cr(III). [4] Standard limit value for Cr(VI). [5] Critical soil metal concentrations for the toxicity effects are likely [43].

### 3.2. Characterization of Amended Mining Soils: Trace Metal Mobility

Tables 4–6 show pH, EC, and mobile metal content of the control soils and soils amended with the four biochars of the three areas at the end of the experiment. The addition of biochars significantly increased the pH of the soils of the Zarandas-Andalusia area (Table 4), shifting the pH from acidic to

alkaline. The pH of the Z1 soil (pH: 5.81) increased by more than 2.2 pH units, while the pH of Z2 soil (pH: 5.10) increased by more than 2.8 pH units; for the Z3 soil (pH: 3.43), biochar addition increased the pH by more than 2.2 units. The addition of biochars significantly increased the pH of the soils of the Portman-Murcia Area (Table 5) although only the pH of the P2 soil (pH: 5.06) changed from acidic to alkaline values. The other three soils had an alkaline pH, with the pH increment after biochar addition being around 1 unit. The effect of biochars in the soil pH in the Mijarojos-Cantabria area was different in the three soils (M1, M2, M3). Initial pH values of soils M1, M2, and M3 were alkaline. There was no significant effect on the pH of M1 soil (pH: 7.91) after biochar addition. In M2 (pH: 8.23), there was an increase in soil pH, while for M3, a liming effect was found only for some biochars. There was a significant increase in electrical conductivity (EC) after biochar addition in the three areas (Tables 4–6). For the Zarandas-Andalusia area (Table 4), the EC of the three soils (EC: 0.09–0.24 dS·m$^{-1}$) increased to values between 1.61 and 3.61 dS·m$^{-1}$; for the Portman-Murcia area (Table 5), the EC of the four soils (EC: 0.37–3.10 dS·m$^{-1}$) increased to values between 1.46 and 5.41 dS·m$^{-1}$. Finally, for the Mijarojos-Cantabria area, EC increased more than 170 times for soil M1, 330 times for M2 soil, and 105 times for soil M3 from an initial EC lower than 1 dS·m$^{-1}$.

**Table 4.** pH, Electrical conductivity (EC), and mobile metal content in control soils and biochar/soil mixtures from Zarandas area after 180 incubation days.

|  | pH | EC | Cr | Cu | Ni | Fe | Zn | Cd | Pb |
|---|---|---|---|---|---|---|---|---|---|
|  |  | dS·m$^{-1}$ |  | mg·kg$^{-1}$ |  | kg·kg$^{-1}$ |  | mg·kg$^{-1}$ |  |
| **Z1** | 5.81a | 0.16a | 0.56b | 0.41b | 6.10c | 5.22d | 1.46b | 0.07a | 6.89d |
| **Z1 + BPL450** | 8.44d | 2.23c | 0.35a | 0.50c | 3.81a | 1.84a | 0.32a | 0.07a | 2.78a |
| **Z1 + BPL600** | 8.46d | 2.24c | 0.31a | 0.65d | 3.74a | 3.48c | 0.32a | 0.06b | 2.28ab |
| **Z1 + BRM450** | 8.07b | 1.61b | 0.73c | 0.61c | 3.98ab | 2.61b | 0.28a | 0.06b | 3.62bc |
| **Z1 + BRM600** | 8.21c | 1.65b | 0.70c | 0.15a | 4.66b | 2.15ab | 0.28a | 0.07a | 4.17c |
| **Z2** | 5.10a | 0.09a | 1.07d | 0.54b | 0.54c | 6.84d | 1.37c | 0.07a | 6.35c |
| **Z2 + BPL450** | 9.43e | 2.29c | 0.23a | 0.21a | 0.29a | 3.10c | 0.13a | 0.06b | 5.93b |
| **Z2 + BPL600** | 9.18d | 2.50d | 0.70c | 0.18a | 0.40b | 1.96a | 0.28b | 0.06b | 5.38b |
| **Z2 + BRM450** | 7.73b | 2.05b | 0.50c | 0.24a | 0.29a | 2.16ab | 0.16a | 0.05b | 3.75a |
| **Z2 + BRM600** | 7.97c | 2.25c | 1.08d | 0.21a | 0.34a | 2.70bc | 0.18a | 0.05b | 4.08a |
| **Z3** | 3.86a [1] | 0.24a | 1.51d | 0.45b | 0.17e | 0.54a | 0.77d | 0.07a | 5.25d |
| **Z3 + BPL450** | 7.29b | 3.49c | 0.78b | 0.15a | 0.46b | 0.34b | 0.22b | 0.05a | 3.57a |
| **Z3 + BPL600** | 7.66c | 3.61c | 0.46a | 0.18a | 0.63c | 0.31b | 0.14a | 0.05b | 4.80c |
| **Z3 + BRM450** | 8.13d | 2.94b | 0.79b | 0.11a | 0.01a | 0.41b | 0.40c | 0.06b | 4.11ab |
| **Z3 + BRM600** | 8.08e | 2.75b | 1.20c | 0.15a | 0.85d | 0.35b | 0.43c | 0.06b | 4.32bc |

[1] Values in different columns followed by the same letter are not significantly different (*p* = 0.05) using the Duncan test for each soil with their amendments.

**Table 5.** pH, electrical conductivity (EC), and mobile metal content in control soils and biochar/soil mixtures from Portman area after 180 incubation days.

|  | pH | EC | Cr | Cu | Ni | Fe | Zn | Cd | Pb |
|---|---|---|---|---|---|---|---|---|---|
|  |  | dS·m$^{-1}$ |  | mg·kg$^{-1}$ |  | kg·kg$^{-1}$ |  | mg·kg$^{-1}$ |  |
| **P1** | 8.42a [1] | 0.37a | 0.24a | 0.38a | 1.22d | 11.69b | 0.21a | 0.83a | 14.24b |
| **P1 + BPL450** | 8.63b | 2.21c | 0.11a | 0.38a | 1.67e | 15.46c | 0.25a | <0.01b | 12.09ab |
| **P1 + BPL600** | 8.82c | 2.35c | 0.42b | 0.32a | 1.07c | 7.29a | 0.51b | <0.01b | 5.72a |
| **P1 + BRM450** | 8.63b | 2.00bc | 0.10a | 0.30a | 0.15a | 9.90b | 0.18a | <0.01b | 5.84a |
| **P1 + BRM600** | 8.72d | 1.46b | 0.08a | 0.42a | 0.91b | 14.41a | 0.28ab | <0.01b | 6.07a |
| **P2** | 5.06a | 1.21a | <0.01b | 0.30ab | 0.59c | 8.18a | 0.37ab | 2.84c | 6.49a |
| **P2 + BPL450** | 7.50b | 3.43c | 0.27a | 0.22a | 1.60e | 8.04a | 0.86c | <0.01b | 4.76a |
| **P2 + BPL600** | 7.86c | 2.93bc | <0.01b | 0.46bc | 0.84d | 8.40a | 0.26a | 0.75ab | 5.66a |
| **P2 + BRM450** | 7.76bc | 2.42b | <0.01b | 0.49c | 0.32b | 4.79a | 0.56b | 0.49a | 5.22a |
| **P2 + BRM600** | 7.60bc | 2.57b | <0.01b | 0.46bc | 0.01a | 12.53b | 0.30a | 1.50b | 5.67a |
| **P3** | 7.69a | 2.28a | <0.01a | 0.14ab | <0.01c | 1.74a | 0.45c | 0.79a | 4.74ab |
| **P3 + BPL450** | 8.65d | 3.95b | <0.01a | 0.23bc | <0.01c | 19.01c | 0.41c | 1.53ab | 8.23c |
| **P3 + BPL600** | 8.19c | 5.21b | <0.01a | 0.10a | 1.29b | 6.00b | 0.06a | 1.54ab | 9.74c |
| **P3 + BRM450** | 7.91ab | 3.70ab | <0.01a | 0.27c | 0.99a | 17.20c | 0.15a | 2.27b | 3.55ab |
| **P3 + BRM600** | 8.03bc | 3.92b | <0.01a | 0.29c | 0.81a | 4.98ab | 0.10a | 1.30ab | 1.29a |
| **P4** | 7.57a | 3.10a | <0.01a | 0.24b | 1.24bc | 8.28a | 0.16a | 0.47a | 3.62b |
| **P4 + BPL450** | 8.02b | 5.41c | <0.01a | 0.20ab | 1.47c | 9.00ab | 0.36b | 1.18bc | 8.17c |
| **P4 + BPL600** | 8.22bc | 5.36c | <0.01a | 0.18a | 0.19a | 7.50a | 0.30b | 0.42a | <0.01d |
| **P4 + BRM450** | 8.59c | 4.48bc | <0.01a | 0.38c | 0.08a | 14.91b | 0.56c | 1.45c | 2.50a |
| **P4 + BRM600** | 8.10b | 4.10b | <0.01a | 0.28c | 0.96b | 9.69ab | 0.24a | 0.81ab | <0.01d |

[1] Values in different columns followed by the same letter are not significantly different (*p* = 0.05) using the Duncan test for each soil with their amendments.

**Table 6.** pH, electrical conductivity (EC), and mobile metal content in control soils and biochar/soil mixtures from Mijarojos area after 180 incubation days.

| | pH | EC | Cr | Cu | Ni | Fe | Zn | Cd | Pb |
|---|---|---|---|---|---|---|---|---|---|
| | | dS·m$^{-1}$ | | mg·kg$^{-1}$ | | kg·kg$^{-1}$ | | mg·kg$^{-1}$ | |
| **M1** | 7.91a [1] | 0.90a | 0.20a | 0.08ab | 2.98a | 4.20c | 1.97b | 4.93c | 8.35b |
| **M1 + BPL450** | 7.96a | 3.11b | 0.85d | 0.04a | 3.12a | 2.09b | 0.28a | 1.46a | 6.85b |
| **M1 + BPL600** | 7.59a | 3.14b | 0.96d | 0.14ab | 6.56b | 1.99b | 0.43a | 1.10a | 5.19a |
| **M1 + BRM450** | 7.49a | 3.05b | 0.54c | 0.16b | 7.37b | 1.43a | 0.30a | 2.94b | 16.57d |
| **M1 + BRM600** | 7.43a | 2.43b | 0.40b | 0.15b | 5.65b | 1.05a | 0.55a | 2.50ab | 14.38c |
| **M2** | 8.23a | 0.40a | 0.31a | 0.08a | 5.45ab | 3.81b | 3.37b | 0.40a | 6.43b |
| **M2 + BPL450** | 9.30e | 1.80ab | 0.37ab | 0.16b | 5.61ab | 2.74ab | 0.20a | 0.37a | 6.34b |
| **M2 + BPL600** | 8.88b | 1,72ab | 0.39ab | 0.38d | 3.43a | 1.71a | 0.50a | 0.28ab | 4.00a |
| **M2 + BRM450** | 9.04c | 2.13b | 0.61c | 0.19b | 4.39a | 1.58a | 0.16a | 0.23b | 15.73c |
| **M2 + BRM600** | 9.18d | 1.73ab | 0.47b | 0.26c | 7.23b | 3.56b | 0.40a | 0.34a | 15.33c |
| **M3** | 7.24a | 0.53a | 0.31a | 0.12b | 4.87a | 1.55bc | 0.41a | 0.33a | 7.10b |
| **M3 + BPL450** | 8.57b | 2.44d | 0.37a | 0.37c | 3.78a | 1.10ab | 0.32a | 0.26a | 5.95ab |
| **M3 + BPL600** | 8.05ab | 2.70e | 0.39a | 0.09ab | 6.75b | 0.86a | 0.35a | 0.13b | 7.63b |
| **M3 + BRM450** | 7.71ab | 1.58b | 0.61b | 0.05a | 8.80c | 1.64c | 0.30a | 0.21a | 4.86a |
| **M3 + BRM600** | 8.67b | 1.73c | 0.43a | 0.07ab | 7.34b | 2.70d | 0.27a | 0.27a | 7.64b |

[1] Values in different columns followed by the same letter are not significantly different ($p = 0.05$) using the Duncan test for each soil with their amendments.

The application of all biochars reduced, in general, the mobility of trace metals in the three soils (Z1, Z2, and Z3) of the Zarandas-Andalusia area (Table 4). For the Z1 soil, the four biochars reduced the mobility of Ni (24–39%), Zn (78–81%), Cd (14%), and Pb (39–67%). However, this was not always the case for Cu and Cr. In addition, the four biochars reduced the percentage of trace metal mobility in both Z2 and Z3 soils. The mobility reduction for Z2 soil was as follows: for Cr (0–53%), Cu (56–67%), Ni (80–91%), Cd (14–29%), and Pb (7–41%), while that for the Z3 soil was as follows: for Cr (21–70%), Cu (60–76%), Ni (24–43%), Cd (14–29%), and Pb (9–32%).

The trend in metal mobility in the four soils of the Portman-Murcia area (Table 5) after biochar application was slightly different to that in the area of Zarandas-Andalusia. For both P1 and P2 soils, the four biochars reduced the mobility of Pb and Cd. The mobility reduction for the P1 soil was as follows: Pb (15–60%) and Cd (~100%), while that for the P2 soil was as follows: Pb (13–27%) and Cd (47–100%). Nevertheless, there was not a clear trend for the remaining soils and biochars.

With respect to the Mijarojos-Cantabria area (Table 6), biochars reduced the mobility of Zn (72–95%) and Cd (30–78%) in the three soils of this area (M1, M2, and M3) (Table 6). Nevertheless, there was not a positive effect of biochars in the reduction of Cr mobility, and there was not a clear trend for the remaining trace metals as seen in the Portman-Murcia area.

*3.3. CO$_2$ Emissions and qCO$_2$ Quotient in Amended Mining Soils*

Figure 2 shows the cumulative CO$_2$ emissions of the control soils and the soils amended with different biochars of the three studied areas after 180 days of the incubation experiment. The variation of CO$_2$ emissions of different treatments with respect to control soils adopted the following sequences in the different areas:

- For the Zarandas-Andalusia area (Figure 2a): Soil Z1: Z1 (81 mg·100 g$^{-1}$) < Z1 + BPL600 (+69%) < Z1 + BRM600 (125%) < Z1 + BRM450 (232%) < Z1 + BPL450 (+281%); Soil Z2: Z2 (90 mg·100 g$^{-1}$) ≈ Z2 + BPL600 ≈ Z2 + BRM600 ≈ Z2 + BRM450 < Z2 + BPL450 (+145%); Soil Z3: Z3 + BPL600 (−30%) ≈ Z3 + BRM600 (−32%) < Z3 (141 mg·100 g$^{-1}$) < Z3 + BRM450 ≈ Z3 + BPL450 (+34%).

- For the Mijarojos-Cantabria area (Figure 2b): Soil M1: M1 (347 mg·100 g$^{-1}$) ≈ M1 + BRM600 ≈ M1 + BPL600 ≈ M1 + BRM450 < M1 + BPL450 (+26%); Soil M2 (264 mg·100 g$^{-1}$): there was no significant difference among treatments for soil M2; Soil M3: M3 (340 mg·100 g$^{-1}$) ≈ M3 + BRM600 ≤ M3 + BPL600 (+15%) ≈ M3 + BRM450 ≤ M3 + BPL450 (+40%).

- For the Portman-Murcia area (Figure 2c): Soil P1: P1 (279 mg·100 g$^{-1}$) ≈ P1 + BRM600 < P1 + BPL600 (+38%) < P1 + BRM450 (77%) ≈ P1 + BPL450; Soil P2: P2 (261 mg·100 g$^{-1}$) ≈ P2 + BPL600

≈ P2 + BRM450 < P2 + BRM600 (+25%) ≈ P2 + BPL450; Soil P3: P3 (295 mg·100 g$^{-1}$) < P3 + BPL450 (+21%) < P3 + BRM600 (+44%) < P3 + BRM450 (62%) ≈ P3 + BPL600; Soil P4: P4 + BPL600 (−33%) ≤ P4 + BRM600 (−28%) < P4 + BRM450 (−17%) < P4 (291 mg·100 g$^{-1}$) < P4 + BPL450 (+27%).

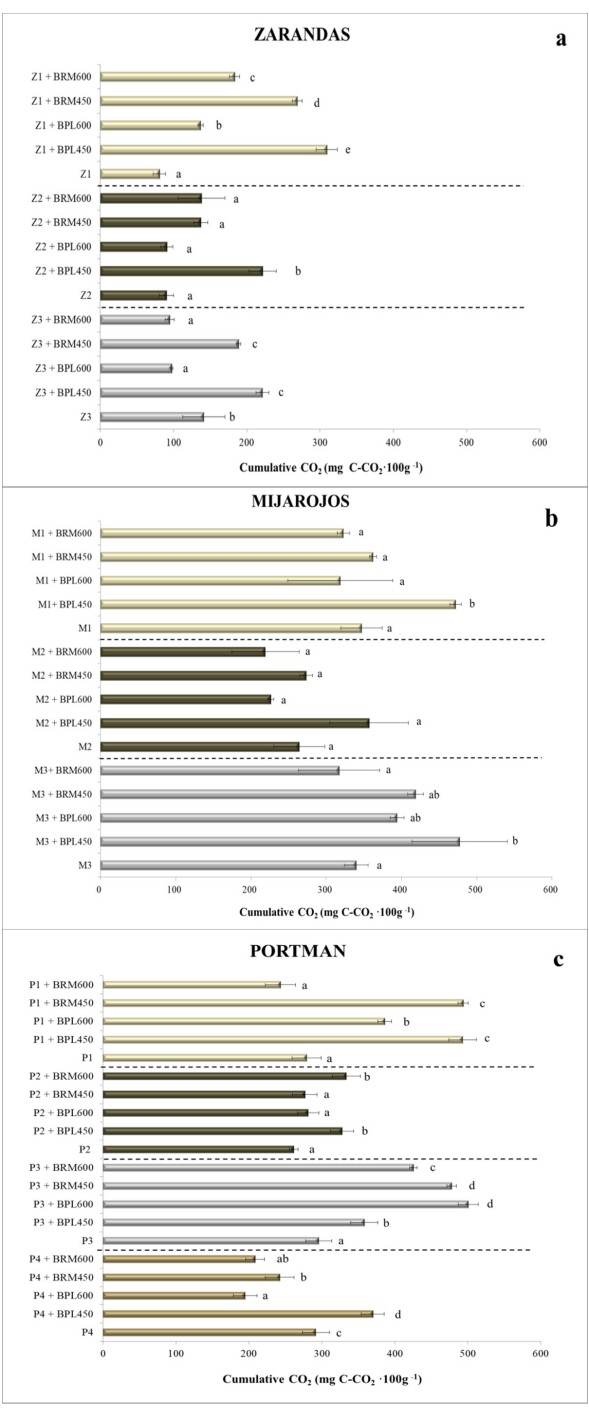

**Figure 2.** Cumulative $CO_2$ emissions of all treatments after 180 days: (**a**) Zarandas-Andalusia area. (**b**) Mijarojos-Cantabria area. (**c**) Portman-area.

The data of $CO_2$ emissions in the Zarandas-Andalusia area showed that the four biochars increased the $CO_2$ emissions in both Z1 and Z2 soils by more than 69% and 145%, respectively. Nevertheless, the application of biochars prepared at 600 °C (BPL600 and BRM600) reduced the $CO_2$ emissions (30–32%) in soil Z3, sequestering C in the soil. In general, the soil $CO_2$ emissions after the application

of biochar prepared at 450 °C were lower than after amendment of biochar prepared at 600 °C. In the case of the Mijarojos-Cantabria area, there was not a significant increment of $CO_2$ emissions in soil M2 after any biochar application. For soil M1, the $CO_2$ emissions only increased after the amendment of BPL450 biochar (+26%), while the $CO_2$ emissions of the soil M3 increased after the application of BPL600 and BRM450 (+15%) and BPL450 (+40%). Finally, for the Portman-Murcia area, the soil $CO_2$ emissions increased by more than 21% in soils P1, P2, and P3 after biochar addition, while the application of BPL600 and BPM600 reduced the $CO_2$ emissions in the soil P4 (−28% to −33%).

Figure 3 shows the soil metabolic quotient (q$CO_2$) values in the three areas after biochar application. Results were different depending on the studied area, although each biochar had a positive effect on q$CO_2$ in more than 60% of the soils. Indeed, BRM600 biochar had a positive effect in seven of 10 soils, improving the microorganism activity in the four soils of the Portman-Murcia area. In general, after the application of biochars, the lowest q$CO_2$ values were generally obtained in the treated soils of the Zarandas-Andalusia area (Figure 3a), while the highest values corresponded to both Mijarojos-Cantabria and Portman-Murcia areas (Figure 3b,c).

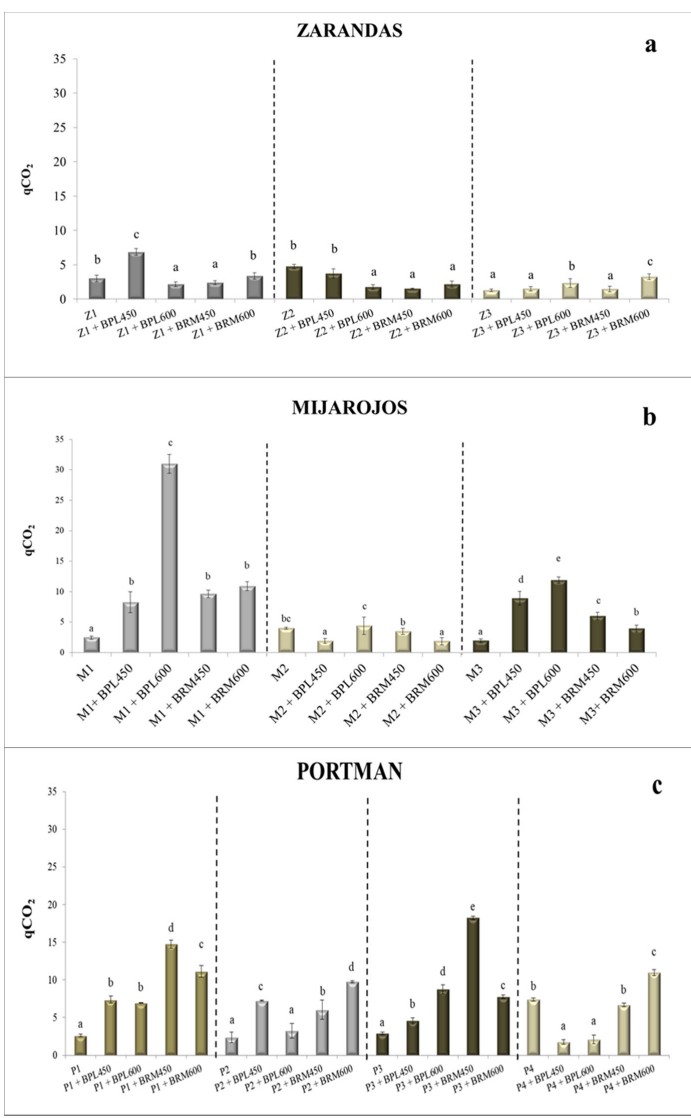

**Figure 3.** q$CO_2$ of all treatments after 180 days: (**a**) Zarandas-Andalusia area. (**b**) Mijarojos-Cantabria area. (**c**) Portman-area.

Figure 4 shows the GMea values for the control soils and soils amended with the four biochars of the studied areas at the end of the experiment. The variation of GMea in different treatments with respect to control soils adopted the following sequences in the different areas:

- For the Zarandas-Andalusia area (Figure 4a): Soil Z1: Z1 + BRM450 (−27%) < Z1 + BPL450 (−35%) < Z1 + BPL600 (−50%) ≈ Z1 + BRM600 < Z1 (0.29); Soil Z2: Z2 + BRM600 (−56%) ≈ Z2 + BPL600 (-61%) < Z2 (0.23) ≈ Z2 + BPL450 ≈ Z2 + BRM450; Soil Z3: Z3 + BPL600 (−18%) < Z3 (0.17) ≈ Z3 + BPL450 < Z3 + BRM450 (40%) ≈ Z3 + BRM600.

- For the Mijarojos-Cantabria area (Figure 4b): Soil M1: M1 + BRM600 (−35%) ≈ M1+BRM450 ≈ M1 + BPL600 < M1 (0.69) < M1 + BPL450 (23%); Soil M2: M2 + BPL600 (−37%) < M2 + BRM600 (−18%) ≈ M2 + BPL450 ≈ M3 + BRM450 < M2 (1.08); Soil M3: M3 + BRM600 (−35%) ≈ M3 + BRM450 ≈ M3 + BPL600 < M3 + BPL450 (−25%) < M3 (0.69).

- For the Portman-Murcia area (Figure 4c): Soil P1: P1 (0.03) ≈ P1 + BRM600 ≈ P1 + BPL600 < P1 + BRM450 (62%) < P1 + BPL450 (323%); Soil P2: P2 + BRM600 (−57%) < P2 + BPL600 (−42%) ≤ P2 (0.07) < P2 + BRM450 (+11%) < P2 + BPL450 (+239%); Soil P3: P3 + BPL600 (−50%) ≈ P3 + BRM600 < P3 (0.04) < P3 + BRM450 (60%) > P3 + BPL450 (264%); Soil P4: P4 + BRM450 (−92%) ≈ P4 + BRM600 < P4 + BPL450 (−50%) < P4 + BPL600 < P4 (0.04).

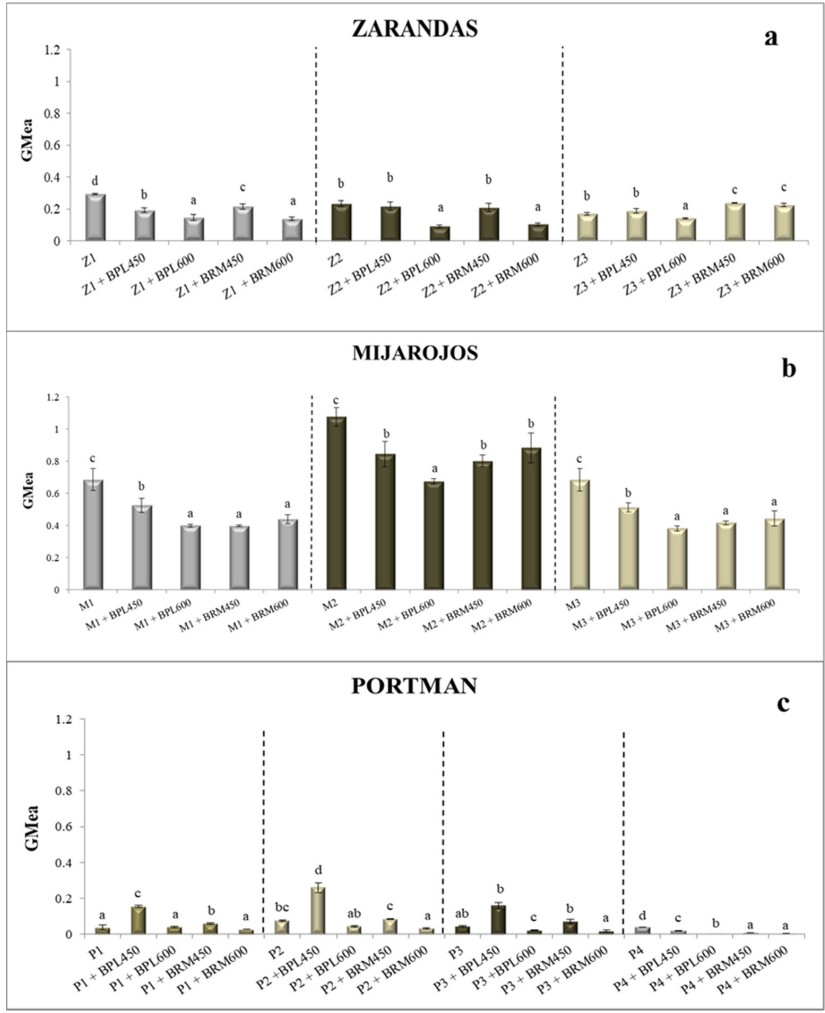

**Figure 4.** Geometric mean (GMea) of all samples after 180 days: (**a**) Zarandas-Andalusia area. (**b**) Mijarojos-Cantabria area. (**c**) Portman-area.

These data showed that the GMea behavior was greatly variable depending on the soil characteristics and biochar added. The control soil of the Mijarojos-Cantabria area (Figure 4b) had higher GMea values than the Zarandas-Andalusia area (Figure 4a) and the Portman-Murcia area (Figure 4c). For the Zarandas-Andalusia area, the GMea index had only a significant increase (+40%) after the application of BRM450 and BRM600 biochars to Z3 soil; for the Mijarojos-Cantabria area, there was only a significant increment for treatment M1 + BPL450 (+23%). Finally, for the Portman-Murcia area, GMea increased in soils P1, P2, and P3 after the addition of BPL450 and BRM450 biochars. These increments ranged from 11% to 264% depending on the soil and biochar.

## 4. Discussion

### 4.1. Materials Characterization

#### 4.1.1. Raw Materials and Biochar Characterization

In general, pH increased with pyrolysis temperature due to ash enrichment and the decarboxylation reactions that take place during thermal treatment [44]. EC values of biochars were slightly higher than those of manure wastes due to ash concentration during pyrolysis. The C-oxi content of manure wastes decreased with pyrolysis treatment, especially at the highest temperature (600 °C) due to the formation of larger and more stable carbon structures that were less oxidized by dichromate [44].

The evolution of CEC during pyrolysis greatly depends on feedstock characteristics. However, in all cases, biochars and manure wastes showed CEC values higher than soil CEC. High CEC values in biochars are recommended for cationic trace element soil remediation [45,46].

Pyrolysis decreased the P-Olsen content, especially in biochars obtained after pyrolysis of RM (BRM450 and BRM600). In general, all P-Olsen biochar values were higher (751–3296 mg·kg$^{-1}$) than those considered as optimum available P for crops (>50 mg·kg$^{-1}$) [47]. Experimental results were similar to those obtained previously by Cantrell et al. [48] and Adhikari et al. [49], where the available phosphorus content decreased in biochars prepared between 300 and 700 °C compared to original wastes. It is proposed that, with increasing pyrolysis temperature, the P availability decreases [50] due to its association with the inorganic fraction of the biochar. Pyrolysis conversion of manure wastes could minimize the adverse impact of manure land application due to phosphorus losses [51–53].

#### 4.1.2. Soils Characterization

In general, all P-Olsen soil values were higher than those in previous agronomic studies (100–269 mg·kg$^{-1}$) [54–56], being considered as available P amount above the optimum for crops (>50 mg·kg$^{-1}$) [47]. Soil metal content of the different areas was very diverse due to the type of mining area. Some areas had important amounts of As, Pb, Zn, Cd, and Cu, being above the critical soil metal concentrations for toxicity effects, according to Kabata and Kabata-Pendias (Table 3) [43] (As: 20–50 mg·kg$^{-1}$; Pb: 100–400 mg·kg$^{-1}$; Zn: 70–400 mg·kg$^{-1}$; Cd: 3–8 mg·kg$^{-1}$; Cu: 60–125 mg·kg$^{-1}$). Specifically, the three areas could have toxic metal content for As, Pb, and Zn (except for the M2 soil that is classified as *Cambisol*), the areas of Portman-Murcia and Mijarojos-Cantabria (soil M1) for Cd, and the area of Zarandas-Andalusia and Portman-Murcia (P2 soil) for Cu. On the other hand, according to the regulations of contaminated soils of the different regions of Spain (see Table 3), the metal soil contents of the Zarandas-Andalusia area were above the regulatory limit values for As (40 mg·kg$^{-1}$ for industrial use), the soils of Portman-Murcia area were above the regulatory limit values for Pb (limit value: 2750 mg·kg$^{-1}$ for industrial use) and As (limit value: 40 mg·kg$^{-1}$ for industrial use), and the soils of the Mijarojos-Cantabria area were above the regulatory limit values for Cd (limit value: 1 mg·kg$^{-1}$), Pb (limit value: 58 mg·kg$^{-1}$), and As (M1 and M3 soils are *Technosols* located in mining tailings). According to these data, the use of biochar in these types of soils could be justified due to the reduction of mobile and available trace metal forms, reducing the toxic effect on soil microorganisms and plants [38].

### 4.2. Metal Mobility in Amended Mining Soils

The application of all biochars reduced, in general, the mobility of trace metals in the three soils (Z1, Z2, and Z3) of the Zarandas-Andalusia area. This can be related to the addition of biochars with high pH, leading to a liming effect in acid soils, decreasing metal mobility [22]. Paz Ferreiro et al. [57] indicated that biochar acts on the heavy metal available fraction, thus achieving a reduction in their mobility in the soil solution. The reduction of the Pb mobility in the Zarandas-Andalusia area could be due to Pb precipitating at soil pH 4–8 and its mobility being reduced in alkaline conditions [58]. Furthermore, previous studies observed that biochar addition raised the pH and increased the soil organic matter content, which favors the immobilization of some metals in acid polluted soils [59,60]. It is important to note that the Z1 soil showed the highest mobile Zn content (1.46 mg·kg$^{-1}$) (Table 4). It is well known that Cu and Zn adsorption can be reduced by the competition between both metals for the same adsorption sites [61] and, consequently, the higher Zn content could explain the increase in Cu mobility in some cases after biochar addition. Moreover, metal adsorption on biochar will depend not only on soil and biochar pH, but also on surface functional groups and pore size distribution [56] (see data in Table 1). In our research, the highest effect of biochar on metal mobility was obtained in soils from the Zarandas-Andalusia area characterized by acid pH and, in the particular cases of Z2 and Z3, low Coxi content. This is in agreement with Al-Wabel et al. [62], who indicated that Cu tends to the formation of organic complexes with organic matter, which facilitates the reduction of its mobility. Therefore, the application of manure biochars to this type of multi-contaminated acid soil can be an adequate strategy to reduce the mobility of trace metals and, therefore, their possible toxicity of metals for soil microorganisms and plants. In the case of the Portman-Murcia area, the main result was the reduction of Pb and Cd mobility in P1 and P2 soils, while, in the Mijarojos-Cantabria area, the main result was the reduction of Zn and Cd mobility due to the causes mentioned above. Indeed, the alkaline pH of the soils of these two areas, except P2 soil, could be the main factor that limits the reduction of trace metal mobility in these two areas.

### 4.3. CO$_2$ Emissions, qCO$_2$ Quotient, and GMea Index

The CO$_2$ emissions of soil after the application of biochars depends on different factors. For example, Cely et al. [21] identified diverse biochar properties that can have an influence on the soil carbon mineralization after biochar addition to soil, including labile carbon content, metal biochar content, or surface biochar properties. For the Zarandas-Andalusia area, the application of the four biochars increased the CO$_2$ emissions in Z1 and Z2 soils, being lower in soils treated by biochar prepared at lower temperatures due to their lower C-oxi and SOC. Moreover, the C sequestration in soil Z3 after the application of biochars BPL600 and BRM600 could be due to part of the CO$_2$ evolved being fixed on the surface area of biochars BPL600 and BRM600 (Table 1). For the Mijarojos-Cantabria area, in general, the application of biochar did not increase the CO$_2$ emissions, which could be associated with the reduction of the microorganism activity in this area after biochar addition (see GMea index in Figure 4). Finally, the increment of CO$_2$ emissions in the majority of the treatments from the Portman-Murcia area could be related to the factors mentioned above with the sandy texture of the soils in this area.

qCO$_2$ is generally used to evaluate how the microbial biomass is using the available carbon for its biosynthesis in an efficient way [63]. According to Odum et al. [64], high qCO$_2$ values are indicative of microbial populations diverting the energy destined for growth and production toward their maintenance and reparation of damages caused by disturbances in the environment. There was, in general, a reduction of the qCO$_2$ parameter in the area of Mijarojos-Zarandas due to the important pH variations in Zarandas soils after biochar addition, which could greatly disturb the environment soil microbial conditions. According to Zheng et al. [65], this may indicate three different situations: a possible degradation efficiency from the microbial community, a possible ability of biochar to protect microorganisms against disturbances or stress, and a possible low microbial activity due to the presence of recalcitrant carbon. Therefore, biochar amendments have different effects on the soil respiration and qCO$_2$ depending on the characteristics of soil and biochar used.

GMea is an index that was used in previous works to measure changes in soil quality after biochar addition to soils [38,66,67]. In this study, GMea behavior was greatly variable depending on the soil characteristics and biochar added. Similarly, a previous study found a contrasting effect of P-rich biochars on GMea [68]. According to Paz-Ferreiro et al. [38], the reduction in GMea index after the addition of biochar could be due to the absorption of substrates and enzymes on the biochar surface. The most positive results of GMea index were found for three soils (P1, P2, and P3) of the Portman-Murcia area where GMea index increased after the addition of BPM450 and BPL450. This fact was probably due to the higher content in labile carbon according to both organic carbon (C-oxi) and soluble organic carbon (SOC) contents and P-Olsen of the biochars prepared at lower temperature [35].

## 5. Conclusions

The application of manure biochar was identified in this study as an adequate strategy for the recovery of mining soils within a circular economy framework with environmental implications. The pyrolysis of manure is a way of waste valorization, especially in countries with a high livestock load such as Spain, where more than 120,000,000 t per year of manure are produced. Moreover, manure biochars can reduce trace metal mobility in mining soils, being also a good strategy for soil carbon sequestration.

Specifically, the application of manure biochars prepared at pyrolysis temperatures between 450 and 600 °C at a rate of 10% can be a good strategy for the recovery of multi-contaminated acid mining soils. Of particular importance would be trace metal immobilization, as it could reduce the phytotoxic effects in plants. This fact, together with the input of nutrients through biochar addition, could have positive effects on vegetation implantation in mining soils, favoring their restoration.

**Author Contributions:** All authors contributed to the intellectual input and provided assistance in the study and manuscript preparation. Conceptualization, M.L.Á., A.M., J.P.-F., and G.G.; formal analysis, M.L.Á., A.M., J.P.-F., and G.G.; writing—original draft, M.L.Á., A.M., J.P.-F., and G.G. All authors have read and agreed to the published version of the manuscript.

**Funding:** This research was funded by the Ministerio de Economía y Competitividad (Spain), grant number CGL2014-58322-R.

**Conflicts of Interest:** The authors declare no conflicts of interest.

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
