# Peer review of "Effects of Manure Waste Biochars in Mining Soils"

_applsci, doi:10.3390/app10103393_

Round 1
Reviewer 1 Report
Presented manuscript describes the potential use of manure derived biochar in reducing risk of heavy metal pollution presenting nice set of data from different mining sites in Spain. Recently a lot of attention has been paid to biochar application to soil for reclamation purposes, however the knowledge about “animal” derived biochar is less studied, what makes this research valuable. Unfortunately authors wrote this manuscript carelessly from the “Introduction” to missing “Discussion” part. The title is “Remediation of mining soils by addition of manure waste biochar” and described research is not reflecting the title. I have some doubts, because authors wrote that the soil and biochar samples were digested with concentrated acids, there is nothing about extraction of mobile/available forms of HM in tested soil which are presented in tables 3,4,5. In table 1. This is confusing, reflecting whole manuscript understanding. In my opinion this paper needs major changes,at first needs to be rewritten to fulfill the requirements of a Applied Sciences journal, also manuscript needs English language check.
Some tips how to improve this manuscript:
- Rewrite the Abstract it’s a copy-paste from the different parts of the manuscript, the most important conclusions of this research are not presented, so it’s quite boring and “catchy” for the potential reader
- There is a lot of typos starting from an Abstract line 16 “on soil nd biochar” line 80 some strange unit 3°C min-1 through whole manuscript
- The Introduction part, nothing novel and not really introducing to the work of Authors and results e.g. lines 61-63 “manure-derived can be used as organic amendment due to high content of nutrients” but nothing about nutrient content in studied biochars, expect available phosphorus is mentioned in the paper. Writing about material for soil remediation Authors should provide more data about total carbon, total nitrogen, nutrient or mineral ash content in biochar. Make a comment on that.
- Separate Results from Discussion, develop Discussion, there is really a lot of research papers to discuss with, focus on your results
- Results are very chaotic, hard to catch the Authors. Why only some regions from described have law regulations about HM content? Something is missing between lines 139-144, what is about sentence in line 139? Rewrite paragraph about regulatory limits lines 144-152, totally incomprehensible for the reader. Try to put more data in text to describe results, to many “up” and “down”, “increase” and “decrease” hard to catch.
Line 166 what the Authors meant saying “CEC values higher than soil CEC values recommended for soil remediation”
- All tables are really big, hard to read them and follow the text looking at so many numbers and significance “a,b,c,d,e,f….till h” so many letters, hard to find out is there a difference or not. Tables need to be shorter
- I was trying to catch with Figure 1. there is too many data on each…
- Conclusions to long and to detailed
Reviewer 2 Report
Comments are in the document attached

Reviewer 3 Report
In this work, the authors deal with the synthesis and application of four different types of biochar in 10 soil samples. Biochar were obtained from rabbit and poultry manure at 450°C and 600°C. Soil samples were taken in three Spanish mining areas. Authors studied the variation of available metal(loid)s concentration and CO2 emission after amendments application. However, several errors were detected, thus the manuscript should be rejected.
The mainly errors is the structure of the article and the statistical method approach, which should be correctly applied in order to explain the results. It is described in detail in the following advices. Furthermore, the biochar characterization could be improved in order to enrich the work, increasing the scientific interest.
Introduction
Line 40: Impermeable barrier or permeable barrier? The reference 6 is about barrier technologies, as well as stabilization/solidification.
The difference between classical and new technologies are relative and subjective. Phytoremediation could be considered a classical technique, although the advances in the last years modified the technique. Other example, permeable barriers formed by clay are classical technologies, although the injection of nanoparticles in an aquifer making a permeable reactive barrier is a new technology. In my opinion, second paragraph should be improved.
Materials and methods
Line 77: Why did you sieved below 5 mm the raw material, and then sieved the biochar at 2 mm?
Line 95: I suggest the use of other abbreviations, for example only P or P-Olsen, and C or C-oxi.
Line 101: Why did you use the dose of 10%?
Line 103: Did you add an extra dose of 10% of biochar or the 10% of the treated-sample is biochar? 50 g of sample is composed by 45 g of soil and 5 g of biochar?
Line 108: reference with different format.
Line 111: Did you use standard solutions for calibration?
How did you determine total metal concentration in soils? What type of extraction did you use? If you use Aqua Regia, you are determining the pseudo-total concentration, because it is not a complete digestion.
Results
This section should be Results and discussion.
Line 134: Zarandas soil samples are contaminated by As, but the As concentration is not shown in table 1.
Line 137: Cr concentration should not be compared with CrVI if you did not know the metal speciation.
Line 145: There are soil screening levels for metal(loid)s for Cantabria (see next reference).
Iribarren, J. Locutura, A. Bel-lan, S. Martinez, J. Grima, S. López, L. Rodríguez, A. Fernández. Determinación de niveles de fondo y niveles genéricos de referencia para protección de la salud humana de metales pesados y otros elementos traza en suelos de la Comunidad Autónoma de Cantabria. Instituto Geológico y Minero - Consejeria de Medio Ambiente del Gobierno de Cantabria (2011).
I found this reference in this other article, which shown the Risk-Based Soil Screening Levels in table 1:
Boente, I. Martín-Méndez, A. Bel-Lán, J.R. Gallego. A novel and synergistic geostatistical approach to identify sources and cores of Potentially Toxic Elements in soils: An application in the regios of Cantabria (Northern Spain).
If you are comparing the (pseudo-)total concentration with the Soil Screening Levels in a mining area, it is more correct to use the industrial use reference. In a mining area, the geochemical background is higher than in other areas due to the presence of a mineral paragenesis. Therefore, the metal(loid)s concentration should be higher due to geological processes, although mining activities could increase these concentrations.
Table 1: The nomenclature should be improved. It is not easy to read the text and follow the data. I suggest to simplify the nomenclature, for example:
Zarandas: Z1, Z2, Z3
Portman: P1, P2, P3, P4
Mijarojos: M1, M2, M3
Line 157-159: I agree with the next affirmation: when pyrolosis temperature is increase, the decarboxylation reactions increase the pH. However, why the ash enrichment increase pH? Are you referring to CEC increase? Several references are needed here. Furthermore, a bivariate correlation (Pearson for example) could be performed between pH and CEC, it should be significant.
Table 2: The statistical treatment performed by ANOVA is not appropriate. In my opinion, several ANOVAS should be performed: two anovas, for each raw material and biochar, in order to explained if the pyrolysis modify raw materials properties; other anova between the soil samples to determine the differences between each other.
In line 57-58, authors claimed: The effect of biochar in soils depends greatly on the biochar characteristics, which are influenced by pyrolysis conditions and feedstock characteristics. For this reason, an exhaustive biochar characterization is essential to know the biochar effects in a particular soil
I totally agree with this affirmation. Therefore, I suggest to improve the characterization of the biochar used in this works (see next).
Line 191: That is true, it is in accordance with lines 57-58, thus it is possible to determine the difference between the pore sizes by electron microscopy, surface area by ASSAP, and surface functional groups by infrared spectroscopy. I recommend to use these techniques to compare both biochars.
Table 3: The statistical treatment should be correct. The ANOVA with all data it is not clear and useful. It should be performed by soils or by treatments, in order to explained the difference between the biochars and the effects in function of soil properties.
Finally, several format errors were detected:
Line 16: nd is and.
Line 21 and more times along the manuscript: CO2 should be written as CO2.
Line 50: metals instead of metal. Or it is possible to use metal(loid)s.
Line 81: should be use subscript and superscript in N2 and -1.
Line 84: the correct symbol is °C.
Reviewer 4 Report
Dear authors,
thank you very much for very interesting reading and the topic of the research is important. Before acceptance of the manuscript, I have some comments and questions that can may potentially improve the quality of the paper.
In Abstract part, I would explain in brackets GMea term. Reader will find it later within the text, but it is important to mention the explanation when it appears for the first time.
I would suggest to change Keywords, as they should not copy the words in the title.
In the intruduction (and also in the conclusion) part, according to the main objective of the work is not the soil remediation (even you mentioned this). Therefore, I would suggest to change the title of the paper.
In Material and methods part I would suggest to add the map of selected minig soils and their location within Spain regions. I also miss detailed description of the collecting localities: source of contaminations, soil types, soil structure, etc. Mostly, soils with different parameters, types, structure cannot be comparable and they might act differently. Because of this, these information are important and must be discussed.
Do you have in Spain some limit values for the soil regarding to the heavy metals in general? You mention some limit values, but it would be simply to follow, when you add them to the table 1 (maybe below as a limit values for each element).
In Results part you state that Zarandas soils are contaminated by As, but I can not see whether you measured this parameter. Can you explain this?
Did you also do analysis of heavy metals for amendments/manures? In Material and methods part you mentioned, that you did this also, so please add this information as well.
Why did you measure total metal content before the experiment and why not the mobility of heavy metals? I think, that when we talk about the remediation and risk potential to the living form, this form is crucial. In table 3-5 you show the mobile form of the heavy metals. Can you compare that with the values you show before the experiment (total heavy metals content)?
Please, also move Figure 2 and 3 within Result part, not after Conclusion part.
line 301- According to (60)....I think you miss to put the authors of this observation
line 315- lesser is not correct english form...use rather lower.
Check the spelling, I have found few mistakes. I have also found few mistakes in the Reference section, so please check the Instruction for the authors and correct it.
I wish you good luck with the publishing of this paper and with your future work.
Round 2
Reviewer 1 Report
Authors responded to all my comments, rewrote the manuscript in most parts. Improve data presentation and description making manuscript more clear and easier to read. Authors discussed findings of the research in a brief way including all relevant points of obtained results. Conclusions are included and clear, I would just erase points 1 and 2 and rewrite as a two sentence.
Author Response
Authors responded to all my comments, rewrote the manuscript in most parts. Improve data presentation and description making manuscript more clear and easier to read. Authors discussed findings of the research in a brief way including all relevant points of obtained results. Conclusions are included and clear, I would just erase points 1 and 2 and rewrite as a two sentence.
Response: Thank you very much for your positive comments and your help for improving the manuscript.
We have erased the points 1 and 2 and we have rewritten the sentences as follows: The pyrolysis of manure is a way of waste valorisation, especially in countries with a high livestock load as Spain where more than 120.000.000 t per year of manure are produced. Moreover, manure biochars can reduce trace metal mobility in mining soils, being also a good strategy for soil carbon sequestration.”
Reviewer 3 Report
Authors made a great effort to improve the work quality and to answer all the reviewers concerns. The statistical analyses and the presentation of the results are now correct. Furthermore, characterization data were included in order to improve the discussion section. In my opinion the manuscript reveals the quality for be published in this journal, although few things should be considered prior its publication.
- Materials and methods. Line 101: Why did you use the dose of 10%?
Thank you for the question. We used this rate due to previous works have shown that can be effective for the treatment of soils contaminated by metals or mining soils. See for example:
- Cárdenas- Aguiar, G. Gascó, J. Paz-Ferreiro, A. Méndez. 2017. The effect of biochar and compost from urban organic waste on plant biomass and properties of an artificially copper polluted soil. International Biodeterioration & Biodegradation 124: 223-232
- Cárdenas‑Aguiar, B. Ruiz, E. Fuente, G. Gascó, A. Méndez. 2019. Improving mining soil phytoremediation with sinapis alba by sddition of hydrochars and biochar from manure wastes. Waste and Biomass Valorization 2020.
These references should be added to justify the applied dose if it is possible.
- Line 134: Zarandas soil samples are contaminated by As, but the As concentration is not shown in table 1.
Thank you for your observation. It was a mistake. We have added the As content of the 10 soils in the new version and also, we have explained in the text that all soils were contaminated by As.
The authors added the As concentration, but what is about the available As concentration? Why did you not measure As on CaCl2 extracts? As concentration in soils is very high in some soils, thus it could be extracted. Biochars, and other organic amendments such as compost, are able to increase As mobility due to dissolve organic content increase or due to phosphorus competition. It would be very interesting to evaluate if manure biochar affect As availability. If data is not available, I suggest to the authors to explore it in following works.
Author Response
1. Authors made a great effort to improve the work quality and to answer all the reviewers concerns. The statistical analyses and the presentation of the results are now correct. Furthermore, characterization data were included in order to improve the discussion section. In my opinion the manuscript reveals the quality for be published in this journal, although few things should be considered prior its publication.
Response: Thank you very much for your positive comments and your help for improving the manuscript.
2. Materials and methods. Line 101: Why did you use the dose of 10%?
Thank you for the question. We used this rate due to previous works have shown that can be effective for the treatment of soils contaminated by metals or mining soils. See for example:
Cárdenas-Aguiar, G. Gascó, J. Paz-Ferreiro, A. Méndez. 2017. The effect of biochar and compost from urban organic waste on plant biomass and properties of an artificially copper polluted soil. International Biodeterioration & Biodegradation 124: 223-232
Cárdenas‑Aguiar, B. Ruiz, E. Fuente, G. Gascó, A. Méndez. 2019. Improving mining soil phytoremediation with sinapis alba by sddition of hydrochars and biochar from manure wastes. Waste and Biomass Valorization 2020.
These references should be added to justify the applied dose if it is possible.
Response: Thank you for your comment. We have added the next sentence: “The dose of biochar has already been used in previous works [35, 36].”
We have added the references as follows:
[35] Cárdenas-Aguiar, E.; Gascó, G.; Paz-Ferreiro, J.; Méndez, A. The effect of biochar and compost from urban organic waste on plant biomass and properties of an artificially copper polluted soil. Int. Biodeterior. Biodegrad. 2017, 124, 223-232.
[36]Cárdenas‑Aguiar, E; Ruiz, B.; Fuente, E:; Gascó, G.; Méndez, A.. Improving mining soil phytoremediation with Sinapis alba by addition of hydrochars and biochar from manure wastes. Waste Biomass Valorization 2020, 1-14.
3. Line 134: Zarandas soil samples are contaminated by As, but the As concentration is not shown in table 1.
The authors added the As concentration, but what is about the available As concentration? Why did you not measure As on CaCl2 extracts? As concentration in soils is very high in some soils, thus it could be extracted. Biochars, and other organic amendments such as compost, are able to increase As mobility due to dissolve organic content increase or due to phosphorus competition. It would be very interesting to evaluate if manure biochar affect As availability. If data is not available, I suggest to the authors to explore it in following works.
Thank you for your comment. We are totally agree with your indication and we will consider it for future works.
Unfortunately, the data are not available as we measured total As in an external laboratory a few months before to start the experiment to have a wide characterisation of the soils. Nevertheless, during the experiment, we used the Atomic Absorption Equipment that we have in our laboratory that is not equipped to measure As due to budgetary constraints.. Nowadays, we cannot send samples to an external laboratory due to the current situation.
Reviewer 4 Report
Dear authors,
thank you very much for incorporation all comments into the manuscript and answering my questions.
I would still add some more information about the source of pollution into the Material and methods part. It would be interesting information for the readers, so reconsider to add it.
Good luck with your future work.
Author Response
1. Dear authors, thank you very much for incorporation all comments into the manuscript and answering my questions. I would still add some more information about the source of pollution into the Material and methods part. It would be interesting information for the readers, so reconsider to add it.
Good luck with your future work.
Thank you for your comment. We agree with you and we have added the next sentences about the main minerals of the different areas. It can give a general view about the origin of the different metals.
Respone: We have added the next sentences:
For Zarandas-Andalusia area: “This area is located in the Iberian Pyrite Belt, one of the largest metallic sulfide deposits in the world with presence of Fe, Zn, Cu, Pb, Cd, As, Ag, Sn or Au.”
For Mijarojos-Cantabria area: “The main minerals of this area are sphalerite, galena, pyrite and secondary minerals of Zn and Pb.”
For Portman-Murcia area: “The main minerals of this area are sphalerite, galena, pyrite and secondary minerals of Zn and Pb.” The main minerals of this area are sulphides ( sphalerite, galena, pyrite), phyllosilicates and carbonates.”